# Top-down, bottom-up, and history-driven processing of multisensory attentional cues in intellectual disability: An experimental study in virtual reality

Jinwook Kim[1ʘ], Eugene Hwang[1ʘ], Heesook Shin[2], Youn-Hee Gil[2], Jeongmi Lee[1]*

**1** Graduate School of Culture Technology, Korea Advanced Institute of Science and Technology, Daejeon, South Korea, **2** Electronics and Telecommunications Research Institute, Daejeon, South Korea

ʘ These authors contributed equally to this work.
* jeongmi@kaist.ac.kr

**Data Availability Statement:** All data files analysis scripts are available from the OSF database: https://osf.io/f4jc9.

## Abstract

Models of attention demonstrated the existence of top-down, bottom-up, and history-driven attentional mechanisms, controlled by partially segregated networks of brain areas. However, few studies have examined the specific deficits in those attentional mechanisms in intellectual disability within the same experimental setting. The aim of the current study was to specify the attentional deficits in intellectual disability in top-down, bottom-up, and history-driven processing of multisensory stimuli, and gain insight into effective attentional cues that could be utilized in cognitive training programs for intellectual disability. The performance of adults with mild to moderate intellectual disability (n = 20) was compared with that of typically developing controls (n = 20) in a virtual reality visual search task. The type of a spatial cue that could aid search performance was manipulated to be either endogenous or exogenous in different sensory modalities (visual, auditory, tactile). The results identified that attentional deficits in intellectual disability are overall more pronounced in top-down rather than in bottom-up processing, but with different magnitudes across cue types: The auditory or tactile endogenous cues were much less effective than the visual endogenous cue in the intellectual disability group. Moreover, the history-driven processing in intellectual disability was altered, such that a reversed priming effect was observed for immediate repetitions of the same cue type. These results suggest that the impact of intellectual disability on attentional processing is specific to attentional mechanisms and cue types, which has theoretical as well as practical implications for developing effective cognitive training programs for the target population.

## Introduction

Intellectual disability (ID) is a neurodevelopmental disorder that appears early in childhood with deficits in both intellectual functioning and adaptive behavior in conceptual, social and

**Funding:** This work was supported by the Electronics and Telecommunications Research Institute (ETRI) grant to HS and YG, funded by Korean government (20ZH1200, The research of the basic media·contents technologies), and the KAIST grant (G04180005) to JL. The funders had no role in study design, data collection and analysis, decision to publish, or preparation of the manuscript.

**Competing interests:** The authors have declared that no competing interests exist.

practical areas [1, 2]. The overall prevalence of ID in the general population is approximately 1% [3], with high comorbidity rates with other neurodevelopmental conditions, such as autism spectrum disorder (ASD) and attention-deficit/hyperactivity disorder (ADHD). Since ID usually persists throughout a person's lifetime, it results in substantial financial and social cost for education and rehabilitation systems [3–5]. Thus, there is an ever-growing need for cognitive training programs that could aid social participation and independence of people with ID. For developing effective training programs, it is crucial to understand the cognitive characteristics of the target population, such as strengths and weaknesses in information processing stages and sensitivity to different types of stimuli, and to use the knowledge as guiding principles in designing the program. Therefore, in this study, we aim to investigate the cognitive characteristics of ID, especially focusing on *attention* that occurs early at the information selection stage of cognitive processing.

The core function of attention is to efficiently select currently relevant information while suppressing irrelevant information. Traditional cognitive models of attention theorized two separate attentional mechanisms, the 'top-down' system that involves voluntary attentional control determined by the observer's goals, and the 'bottom-up' system that involves involuntary attentional control based on the saliency of physical stimuli [6–8]. The functioning of the top-down system is often examined by using endogenous cues, stimuli that are only *symbolic* of the target location (e.g., an arrow at the center of the screen pointing to a specific direction). The functioning of the bottom-up system, on the other hand, is examined by using exogenous cues, stimuli that directly highlight the target location (e.g., flickering in the upcoming target location). Decades of research have supported the dichotomy between top-down and bottom-up attentional mechanisms, and demonstrated that the two attentional mechanisms are controlled by two partially-segregated networks of brain areas; the top-down system recruits the dorsal frontoparietal network that includes the intraparietal and superior frontal cortices, whereas bottom-up system recruits the ventral frontoparietal network that consists of the temporoparietal and inferior frontal cortices [9–12].

While the notion of top-down and bottom-up attentional mechanisms has received substantial evidence in typically-developing people, it remains unclear whether or how the two attentional mechanisms are differentially affected by ID. Previous studies have reported that individuals with ID show deficits in attentional control, for instance, failures in inhibiting attention to distracting stimuli [13, 14], greater susceptibility to interference on Stroop tasks [15] or tasks that require working memory updating [16]. However, it is unresolved whether the attentional deficits in ID are due to vulnerability in the top-down or bottom-up system, or both. Although several studies demonstrated that impairment in top-down processing is more pronounced in other frequently co-morbid neurodevelopmental conditions such as ASD [17–19] and ADHD [20, 21], the functioning of top-down and bottom-up mechanisms in ID has been less explored and provided inconclusive results [22, 23].

In addition to the traditional top-down and bottom-up dichotomy [24, 25], recent theories posit that the 'selection history' should be considered as the third attentional mechanism. The selection history indicates sources of information for attentional guidance that cannot be explained by current goals or physical saliency, including components such as priming, reward-associations, and statistical learning. For instance, stimuli that had been selected repeatedly [26–28] or previously associated with reward [29–31] are often prioritized, even though they are neither goal-relevant nor physically salient. The medial temporal lobe and the hippocampus are generally accepted to be crucial for controlling selection history-driven processing, since it is closely intertwined with learning and memory functions [25, 32, 33]. Considering that the three attentional mechanisms are controlled in three partially-segregated networks of brain areas, it is critical to elucidate whether the functions of all three attentional mechanisms are impaired in

general in ID, or there are specific impairment and preservation of functions across attentional mechanisms. However, few studies investigated the characteristics of history-driven attentional processing in ID, and there has been no study that directly compared the functioning of the three attentional mechanisms in ID in the same experimental setting.

Another critical gap in knowledge regarding the attentional characteristics of ID is that most studies focused on selection process of stimuli within a *single* sensory modality, predominantly in vision, in highly unnatural conditions. The majority of previous studies investigated how the aspects of visual stimuli presented on a 2-D (two dimensional) screen guide covert attentional selection, while observers were required to fixate on the center of the screen. Much fewer studies investigated the attentional process within other sensory modalities in natural conditions where participants are allowed to move their eyes in a 3-D (three dimensional) environment. This was partly due to limitations in the stimulus-presenting system: Although the traditional computer system enabled precise and systematic presentations of visual stimuli, it was not very suitable for presenting stimuli in other sensory modalities in a realistic 3-D environment. Recent advancements in virtual reality (VR) head-mounted displays (HMDs) could be an excellent solution for this problem [34, 35]. VR HMDs have built-in visual display, speakers, and controllers that can systematically deliver visual, auditory, and tactile stimuli to observers. Moreover, the multisensory stimuli presented in the 3-D environment in VR provide higher ecological validity than the 2-D screen, and also enable much better experimental control of stimuli as compared to the real environment. Harnessing the strengths of VR HMDs, many VR training programs are being developed to improve cognitive, motor, social, and daily life skills of people with intellectual and developmental disabilities [36–38]. VR enables learning in a safe, immersive environment, and the effect of training in VR is reported to be as good as that in the real environment in vocational training [39] and life skills training [40]. Thus, we aim to investigate the effect of multisensory (visual, auditory, and tactile) attentional cues on people with ID in a 3-D VR environment, such that results obtained in the current study could be well generalized and applied to design effective attentional cues in VR training programs for the target population.

The current study addresses two main research questions regarding the attentional characteristics of people with ID. The first research question is examining the differential effects of endogenous, exogenous, and repeatedly primed cues on attentional guidance in ID. Since top-down, bottom-up, and history-driven attentional mechanisms are controlled by partially-segregated networks of brain areas, we hypothesize that the impact of ID would not be identical on each attentional mechanism. In that case, significant interaction between group (ID, typically-developing controls) and cue type (endogenous vs. exogenous, repeated vs. non-repeated) would be observed. Evidence for relative strengths and weaknesses across different attentional mechanisms will shed light on the locus of vulnerability in neurological development in ID, and also provide valuable implications on designing effective attentional cues in training programs for ID. The second research question is to examine the efficiency of attentional selection in ID across different sensory modalities in a 3-D environment with higher ecological validity. For this investigation, we deliver attentional cues in three different sensory modalities (visual, auditory, and tactile) in a VR search task, and compare the behavioral performance (response time and accuracy rate to find the search target) across conditions. If ID influences the attentional process differentially across sensory modalities, we would observe significant interaction between group (ID, typically-developing controls) and sensory modality (visual, auditory, tactile) of the cue. Examining the attentional selection process in ID across different sensory modalities in a more natural 3-D environment would fill in the important gap in knowledge, and provide implications on designing effective attentional cues in VR training programs for ID.

In the experiment, a visual search task combined with the Posner spatial-cueing paradigm [41] was performed inside a 3-D VR environment where eye movements were allowed. On each trial, participants searched for a specific target among distractors, while a cue that gives either useful or useless information about the location (the left or right visual field) of the upcoming target was presented before the search array. By comparing the behavioral performance in trials with informative cues versus non-informative cues, we measured the efficiency of each cue in guiding attentional selection. Importantly, the cue type was manipulated to be either endogenous or exogenous, in order to evaluate the efficiency of top-down and bottom-up attentional mechanisms, respectively. Also, cue stimuli were presented in different sensory modalities to examine cross-modal attentional functions. Finally, the efficiency of history-driven attentional mechanism was measured by the size of the repetition priming effect, which indicates the improvements in behavioral performance when the same type of trial was immediately repeated versus not repeated. Priming effects for the repetitions of cue type, target side, and target location were analyzed, to compare the enhancement effect of repeated features versus spatial locations. For exploratory analyses, we included two different feature dimensions for each sensory modality to observe whether the effects of exogenous cues in guiding attention differed across feature dimensions. Also, feedback type on the accuracy of the response was manipulated across blocks to additionally explore whether getting a different type of feedback influenced the overall search performance. By comparing the pattern of performance of adults with ID with that of typically developing controls in this task, we provide a comprehensive picture of relative efficiency of different types of attentional cues in ID.

## Materials and methods

### Participants

A total of forty adults, consisting of twenty adults with ID (15 males and 5 females) and twenty typically-developing (TD) adults (11 males and 9 females), participated in the study (Table 1). The number of participants was determined a priori by a power analysis (G$^*$power version 3.1.9.2) using parameters corresponding to the main analysis of interest, the three-way interaction between group (ID, TD), sense (visual, auditory, tactile), and origin (endogenous,

**Table 1. Characteristics of participants in the TD and ID groups.**

|  | TD group (n = 20) | ID group (n = 20) |
|---|---|---|
| **Age (years)** |  |  |
| Mean (SD) | 28.1 (5.05) | 30.6 (7.26) |
| Range | 24–44 | 22–50 |
| **Gender, n (%)** |  |  |
| Female | 9 (45) | 5 (25) |
| Male | 11 (55) | 15 (75) |
| **ID level, n (%)** |  |  |
| Mild |  | 17 (85) |
| Moderate |  | 3 (15) |
| **Comorbidity, n (%)** |  |  |
| ASD |  | 2 (10) |
| **Handedness, n (%)** |  |  |
| Right-handed | 17 (85) | 14 (70) |
| Left-handed | 2 (10) | 2 (10) |
| Ambidextrous | 1 (5) | 4 (20) |

exogenous), with .85 power to find a medium-size effect ($\eta_p^2$ = .06) in a mixed ANOVA within-between interaction test (number of groups = 2, number of measurements = 6, alpha = .05). The medium-size effect was used in the power analysis based on the previous studies that compared the effect of attentional cues on behavioral performance of a developmental disability group with that of a typically-developing group, which reported medium to large ($\eta_p^2$ = .06 ~ .24) effect sizes for interaction between group and cue type [18, 19, 23, 42]. The participants ranged in age from 22 to 50 years (M = 30.6, SD = 7.26) for the ID group, and from 24 to 44 years (M = 28.1, SD = 5.05) for the TD group, with no statistical difference in mean age between groups [$t(38)$ = 1.23, $p$ = .225, $d$ = .39]. Participants' dominant hand was measured by Edinburgh Handedness Inventory for both ID (right-handed: 14, left-handed: 2, ambidextrous: 4) and TD groups (right-handed: 17, left-handed: 2, ambidextrous: 1). Gender and handedness compositions were compared between groups using Pearson's chi-squared tests. The results confirmed that the ID and TD groups were not significantly different in terms of gender ($\chi^2$ = 1.758, df = 1, $p$ = .185, $V$ = .210) and handedness ($\chi^2$ = 2.090, df = 2, $p$ = .352, $V$ = .229) compositions.

The ID group was recruited from a state-run vocational training center for developmental disabilities. The inclusion criteria were a diagnosis of mild to moderate intellectual disability with recognized ability of basic verbal communication. All participants were diagnosed and prescreened by an experienced, independent group of clinicians at the training center according to the criteria of the DSM-5 (American Psychiatric Association, 2013). Until the DSM-4, the ID severity levels were based only on the IQ scores. However, the DSM-V abandoned specific IQ cutoffs as a diagnostic criterion, and placed more emphasis on the impairments in conceptual, social, and practical life skill domains (APA, 2013). Thus, each participant's diagnosis and severity level of ID was determined by independent clinicians based on comprehensive evaluation of the scores on standardized tests, including IQ test (K-WAIS), Peabody Picture Vocabulary Test (K-PPVT), Bender Gestalt Test (BGT), Individual Basic Learning Skills Test (K-IBLST), Hand Function Test, Work Sample Test, and also clinical interviews and observations. All ID participants were relatively high-functioning (mild ID: 17, moderate ID: 3), and two of them had comorbid ASD. We did not exclude the ASD-comorbid participants to secure the enough sample size for the ID group, since ID and ASD covary at very high rates (28%~40%) and the covaring rates are even more increased after changes in diagnostic criteria in the DSM-5 [43–45]. People who showed any significant perceptual (e.g., visual, auditory, tactile) or motor deficits in clinical observations or in the Adolescent/Adult Sensory Profile [46] were excluded from participation, since it would directly influence the behavioral performance in our experimental task. We also excluded people who had a clinical history of other neurodevelopmental genetic disorders (e.g. Down syndrome, Williams syndrome), which are known to have distinct characteristic symptoms and features. Finally, people who had a history of behavioral problems (e.g., stereotyped behaviors, difficult, disruptive, or aggressive behavior) were prescreened to ensure safety and completion of participation. The TD group did not show any significant perceptual or motor deficits, had no history of mental or brain diseases, and were recruited from a university (KAIST). All the experimental protocol was in accordance with the Declaration of Helsinki, and was approved by the KAIST Institutional Review Board (KH2019-128). Each participant provided written informed consent, along with written informed guardian consent in the ID group.

## Stimuli and apparatus

In a quiet experiment room, participants performed the experimental task with safety certified commercial head-mounted display (HMD), Oculus Rift S (2560x1440 resolution, 115-degree

field-of-view, 80Hz refresh rate). Participants sat on a chair fixed at a specific location, such that all virtual stimuli were within arm's reach (70 cm) from the sitting position with the virtual reality controller in both hands. The virtual stimuli and the task structure were developed with UNITY 3D (2018.4.2f1) and Oculus Integration (1.38.0). The stimuli presented in the virtual environment consisted of the visual and auditory output from the HMD's built-in display and speakers, and the tactile (vibration) output from the HMD's left and right controllers. While participants performed the experimental task, visual, auditory or tactile stimuli were provided as a spatial cue. Participants were allowed to move their eyes or head to explore the environment from the sitting position, and used their virtual hands for response. Participants' behavioral performance (response time, accuracy) under each experimental condition was recorded.

## Experiment design

We designed our task by combining the Posner cueing task [41], which has been widely used to measure the effect of spatial-cueing on selective attention, and the visual search task, in which participants find a specific target object among distracting stimuli (Fig 1A). In the task, first the target number (randomly selected from 0 to 9 for each trial) to search for is presented in white on the center of the black 3-D background, while the verbal sound of the number is also presented through the speakers on both sides (2s). Then, eight (2 x 4) yellow cubes (edge length: 12.1˚) appear, with four cubes located on each side (left, right) of the screen. The distance from the center of the screen to the nearest cube was 10.6˚, and the gap between cubes in

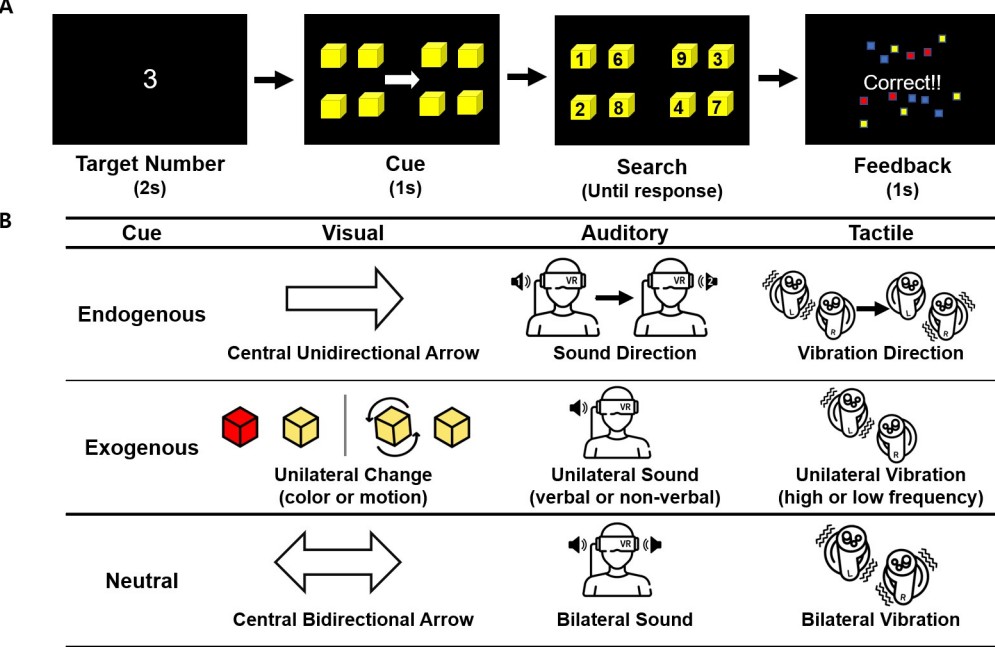

**Fig 1. Illustration of the experimental design.** (A) Sequence of events and time course of a trial in the visual search task. On each trial, a target number to search for was first presented (2s), then a spatial cue that gives information about the location (left or right side) of the upcoming target was presented (1s). In this example, the visual endogenous cue (central unidirectional arrow) is illustrated. Then, participants searched for the cube on which the target number is written and touched it with a virtual hand. The feedback about the accuracy of the response was given (1s). (B) Illustration of different cue types in the experiment. The cue types were categorized based on its sensory modality (visual, auditory, tactile) and origin (endogenous, exogenous). Neutral cues did not give any information about the location of the upcoming target.

the same side was 12.6˚. The cubes were located in the near-peripheral visual field, rather than in the central visual field, in order to examine the *overt* attentional selection process that occur in a 3-D environment where natural eye and head movements are allowed. After 250 ms of interval, an attentional cue that gives information about on which side (left or right) the target will appear is presented for the duration of 1s. Immediately after the offset of the cue, eight different numbers (randomly selected from 0 to 9, including the target number) appear randomly mapped on each cube. Participants were instructed to respond as quickly and accurately as possible by finding and touching the cube on which the target number is written by controlling a virtual hand, and the response time and accuracy of the response was measured on each trial. Immediately after the response, the cubes disappear and the feedback about the accuracy of the response is given (1s).

The cue types were categorized based on its sensory modality (visual, auditory, tactile) and origin (endogenous, exogenous) to compare the selection process in different sensory modalities and attentional mechanisms (top-down, bottom-up) (Fig 1B). For exogenous cues, two different feature dimensions were included for each sensory modality to additionally explore whether the effects of exogenous cues in guiding attention differed across feature dimensions. First, as a visual endogenous cue, a white arrow appeared on the center of the screen, pointing towards the side (left, right) in which the target would subsequently appear. For visual exogenous cues, the four cubes located on the side in which the target would subsequently appear changed their color to red (color cue), or rotated clockwise (motion cue).

For an auditory endogenous cue, the direction of sound source movement was used to provide information about where the target would appear; The "ta-dak" sound was played starting from the left speaker and ending in the right speaker, if the target would subsequently appear on the right side of the screen, and vice versa. We utilized the direction of sound, instead of language (e.g., the verbal sound of "left" or "right"), in order to make endogenous cues in different sensory modalities as comparable as possible. Since the visual endogenous cue was a nonverbal symbolic sign (arrow direction), we used similar nonverbal directional symbols in auditory and tactile modalities. For auditory exogenous cues, the verbal sound of "this side" (verbal cue) or nonverbal sound of "Ding-dong" (nonverbal cue) was presented only on one side of the speakers that corresponds to the side that will subsequently contain the target.

For a tactile endogenous cue, the direction of vibration was used to provide information about the target location; The vibration was presented starting from the left controller and ending in the right controller, if the target would subsequently appear on the right side of the screen, and vice versa. For tactile exogenous cues, low- (low-frequency cue) or high-frequency vibrations (high-frequency cue) were presented only on one side of the controllers that corresponded to the side of the subsequent target.

Finally, there were neutral cues that did not give any information about where the target will appear. Neutral cues consisted of a central arrow pointing towards both sides (visual), the "Ta-dak" sound played on both sides of speakers at the same time (auditory), and vibrations on both sides of controllers at the same time (tactile). We included neutral cues, instead of 'invalid' cues typically used in the spatial cueing paradigm [41], since probability manipulation between valid and invalid trials should require statistical learning abilities, which are reported to be significantly different between TD and ID groups [47], and therefore would confound the results obtained from the two groups. Since neutral cues were identical in terms of containing no information about the location of the upcoming target, we checked the homogeneity in behavioral performance between neutral cue types by conducting a mixed ANOVA with neutral cue type (visual, auditory, tactile) as a within-subject factor and group as a between-subject factor. The results showed no significant main effect [$F(2, 76) = 1.82$, $p = .17$, $\eta_p^2 = .046$] or

interaction [$F(2, 76) = .28$, $p = .758$, $\eta_p^2 = .007$] involving neutral cue type, and therefore search performance from neutral cue trials were collapsed in subsequent analyses and used as a baseline condition for each participant.

Immediately after each response, feedback on the accuracy of the response was either not given at all, or given by a visual, auditory, or tactile stimulus. The feedback type was manipulated across blocks to additionally explore whether getting a different type of feedback influences the motivation of the participants, and in turn the overall search performance. The visual feedback presented text on the screen that informed whether the response was correct or not. For auditory feedback, a fanfare or a low tone was presented on both speakers to indicate a correct or incorrect response, respectively. For tactile feedback, both controllers vibrated four times for a correct response or once for an incorrect one. The overall structure of the experiment consisted of four feedback blocks (visual, auditory, tactile, and no-feedback), with the order of blocks determined randomly for each participant. Each feedback block consists of 50 trials (10 types of cues x 5 repetitions) in a random order. As a result, each participant performed a total of 200 trials (10 types of cue x 20 repetitions) in the main experiment.

## Procedure

Before starting the main experiment, participants received detailed explanations about the task and the meaning of each cue type. Then they wore the HMD and performed a practice block (20 trials) that consisted of all types of trials in a random order. During the practice block, at least two experienced experimenters constantly monitored each participant's behavioral performance (accuracy rate) and verbally checked the understanding of each cue type. When the participant showed no sign of understanding the cue information, experimenters immediately intervened and explained the rules again. Instructions and practice blocks were repeated until all experimenters agreed on the participant's successful understanding of the meaning of each cue type and a high level of accuracy rate (over 90%). The ID group spent approximately twice as much time on practice blocks as compared to the TD group. After confirming that each participant understood the task and the meaning of each stimulus, the main experimental task (200 trials in total) was conducted. During the experiment, participants were given a break every 25 trials, during which the progress rate information was presented on the screen. Participants were also able to take a rest anytime if they wanted.

## Statistical analysis

Inverse efficiency (IE) values were calculated and analyzed for each participant/cue condition to combine the effects of response time (RT) and accuracy. The IE is the mean RT of each condition divided by the accuracy rate of each condition for each participant, showing the combined effects in conflicting situations where both high speed and high accuracy are required [48]. Thus, higher IE values generally indicate lower search efficiency (longer RTs and/or lower accuracy). SPSS (version 25.0.) was used for all statistical analyses, with the threshold for significance (alpha level) of 0.05. We first compared behavioral performance when the cue contained useful spatial information (informative cues) versus no information (neutral cues) by entering the IE data into a mixed ANOVA with cue type (informative cues, neutral cues) as a within-subject factor and group (ID, TD) as a between-subject factor. To examine the effect of cue type considering the baseline performance of each participant, we then calculated the cue effect index by subtracting the IE value for each cue type from that of the neutral cue, for each participant. Next, we performed mixed ANOVAs with group as a between-subject factor and different cue type (origin, sense, or repetition) as within-subject factors to answer our main research questions. Only if there were significant interactions involving group, separate

repeated-measures ANOVAs for the two groups were conducted. If significant main effects or interactions were observed in repeated-measured ANOVAs, paired t-tests (two-tailed) were performed for post-hoc comparisons, and Bonferroni-corrected p-values were reported for multiple tests. The Greenhouse-Geisser corrected values (denoted by $F_c$) were reported for the comparisons of variables that violated the assumption of equality of variance.

## Results

The overall accuracy was very high, with 99.7% (range: 98.5% to 100%) mean accuracy rate for the TD group and 98.2% (range: 91.5% to 100%) for the ID group. Only the RTs from correct trials were analyzed, and outliers that were more than 2 standard deviations away from the mean of each participant/cue condition were removed. As a result, 5.43% of trials were eliminated in the TD group and 6.35% of trials were eliminated in the ID group. The mean RT was almost twice as long in the ID group (1527 ms) as compared to the TD group (857 ms). Results from preliminary analyses on the RT and accuracy data, as well as RT and accuracy means for each cue type are reported in the S1 File. To examine whether there was the speed-accuracy tradeoff or not, we correlated each cue type condition's mean RT and mean accuracy rate (refer to the S1 File for details). For the TD group, there was a significant *negative* correlation between RT and accuracy rate ($r = -.80$, $p = .002$), which is the opposite of the definition of speed-accuracy trade-off. For the ID group, the correlation between RT and accuracy rate was not significant ($r = .45$, $p = .143$), showing no sign of the speed-accuracy tradeoff. Since there was no evidence of the speed-accuracy tradeoff, IE values were calculated and analyzed for each participant/cue condition to combine the effects of RT and accuracy in all subsequent analyses. In addition, we conducted parallel analyses on log-transformed RTs to make sure that the pattern observed with the IE data can be reproduced with RTs only, and the interaction between group and cue type is not simply an artifact of the perceptual/motor-speed difference between groups. The results confirmed that all major effects observed with the IE data were replicated with the log-transformed RT data (refer to the S1 File). Finally, even though the ID and TD groups were not significantly different in terms of gender compositions, we verified the possibility that gender of the participants might have differentially influenced attentional performance. The main analyses were conducted with gender as a between-subject factor, and the results revealed no significant main effect or interactions involving gender, supporting that gender did not differentially affected the search performance (refer to the S1 File).

### The effect of different cue types

We first examined whether the spatial cues in our experimental task were effective in guiding participants' attention or not by comparing behavioral performance when the cue contained useful spatial information (informative cues) versus no information (neutral cues). The IE data were entered into a mixed ANOVA with cue type (informative cues, neutral cues) as a within-subject factor and group (ID, TD) as a between-subject factor (Fig 2). The results showed a significant main effect of group [$F(1, 38) = 39.91$, $p < .001$, $\eta_p^2 = .512$], with a higher mean IE in the ID group (1653 ms) than in the TD group (1034 ms). This indicates that the ID group had overall slower RTs and lower accuracy rate as compared to the TD group. There was also a main effect of cue type [$F(1, 38) = 131.06$, $p < .001$, $\eta_p^2 = .775$], with a higher mean IE for neutral cues (1511 ms) than for informative cues (1177 ms). Most importantly, the interaction between cue type and group was significant [$F(1, 38) = 12.10$, $p = .001$, $\eta_p^2 = .241$], indicating that the effect of cue type occurred differently for the typically developing controls and individuals with ID. Separate repeated-measures ANOVAs for the two groups revealed that there was

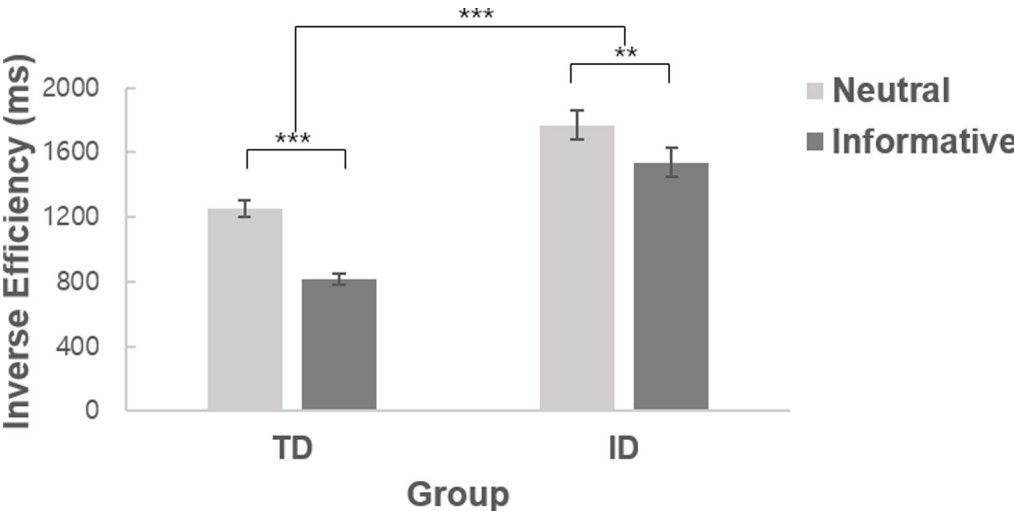

**Fig 2. Mean inverse efficiency for neutral versus informative cues, shown separately for the typically developing group (TD) and the intellectual disability group (ID).** Error bars represent the SEM in all figures. $^{**}p < .01$, $^{***}p < .001$.

a significant main effect of cue type in both the TD group $[F(1, 19) = 130.40, p < .001, \eta_p^2 = .873]$ and ID group $[F(1, 19) = 27.72, p < .001, \eta_p^2 = .593]$, and an independent samples t-test indicated that the difference in mean IE between neutral cues versus informative cues was smaller in the ID group (232 ms) than in the TD group (435 ms) $[t(38) = 3.48, p = .001, d = 1.10]$. These results suggest that even though the informative cues in our experimental paradigm successfully guided participants' attention in general, the amount of benefit from informative cues was relatively smaller in participants with ID than in typically developing controls.

To further examine the effect of cue type considering the baseline performance of each participant, we calculated the cue effect index by subtracting the IE value for each cue type from that of the neutral cue, for each participant. Thus, higher values in the cue effect index represent greater cueing effects. Next, we evaluated our main research question, the efficiency of top-down and bottom-up processing of multisensory cues, by entering the cue effect data into a mixed ANOVA with sense (visual, auditory, tactile) and origin (endogenous, exogenous) of the cue as within-subject factors and group (ID, TD) as a between-subject factor (Fig 3). The results showed a significant main effects of origin $[F(1, 38) = 21.67, p < .001, \eta_p^2 = .363]$ and group $[F(1, 38) = 16.02, p < .001, \eta_p^2 = .297]$, but no significant main effect of sense $[F(2, 76) = 1.80, p = .172, \eta_p^2 = .045]$. Overall, the effect of exogenous cues (368 ms) was greater than that of endogenous cues (264 ms), with a higher mean cue effect in the TD group (431 ms) as compared to the ID group (202 ms). Importantly, the interaction between origin and group was significant $[F(1, 38) = 12.70, p = .001, \eta_p^2 = .251]$, indicating that the effect of origin of cues occurred differently in the TD and ID group. Separate repeated measures ANOVAs for the two groups revealed no significant difference in the cue effect between endogenous and exogenous cues in the TD group $[t(19) = 1.62, p = .122, d = .362]$, whereas in the ID group the mean cue effect was significantly larger for exogenous cues (294 ms) than endogenous cues (110 ms) $(t(19) = 4.37, p < .001, d = .976)$. This result supports the hypothesis that the deficits in ID are more pronounced in the top-down attentional mechanism as compared to the bottom-up attention mechanism. Finally, sense by origin interaction $[F(2, 76) = 19.06, p < .001, \eta_p^2 = .334]$ and sense by origin by group

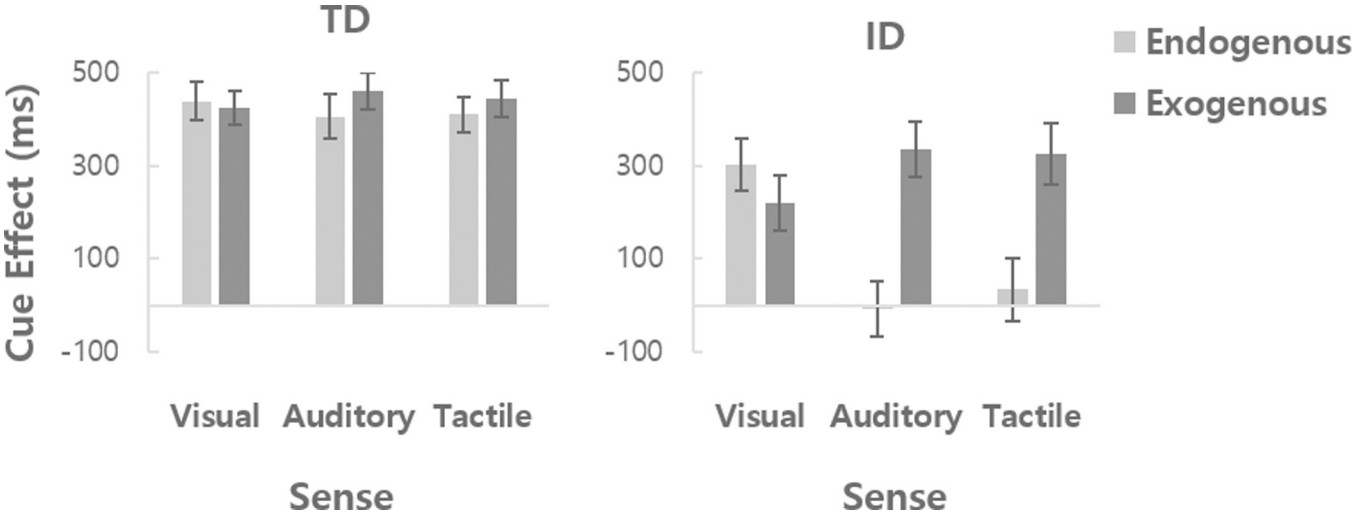

**Fig 3. Mean cue effect for each cue type, based on the origin (endogenous, exogenous) and sense (visual, auditory, tactile) of the cue, shown separately for the TD and ID groups.** The cue effect was calculated by subtracting the IE value for each cue type from that of the neutral cue, such that higher values represent greater cueing effects.

interaction [$F(2, 76) = 10.29$, $p < .001$, $\eta_p^2 = .213$] were significant. Separate repeated measures ANOVAs for the two groups revealed that in the TD group, the main effects of sense [$F(2, 38) = .15$, $p = .861$, $\eta_p^2 = .008$] and origin [$F(1, 19) = 2.62$, $p = .122$, $\eta_p^2 = .121$] were not significant, but the interaction between sense and origin was significant [$F(2, 38) = 3.83$, $p = .03$, $\eta_p^2 = .168$]. Post-hoc comparisons with Bonferroni correction showed no significant difference between exogenous and endogenous cues in all sensory modalities ($t(19)$s $< 2.37$, $ps > .87$, $ds < .529$) in the TD group. In the ID group, the main effect of origin was significant [$F(1, 19) = 19.06$, $p < .001$, $\eta_p^2 = .501$], as well as the interaction between sense and origin [$F(2, 38) = 15.75$, $p < .001$, $\eta_p^2 = .453$]. The post-hoc comparisons with Bonferroni correction showed that exogenous cues were more effective than endogenous cues in both auditory [$t(19) = 6.51$, $p < .001$, $d = 1.455$] and tactile [$t(19) = 3.80$, $p = .003$, $d = .850$] sensory modalities, but not in visual sensory modality [$t(19) = 1.39$, $p = .54$, $d = .311$] in the ID group. Taken together, these results suggest an interesting possibility that even though the function of the top-down attentional mechanism is generally more deteriorated in ID, the level of decline might be cue type-specific: The auditory or tactile endogenous cues were much less effective than the visual endogenous cue in guiding top-down attention of ID.

In the current study, the exogenous cues were manipulated to be salient in six feature dimensions, namely color, motion, verbal, nonverbal, high-frequency, and low-frequency, to additionally explore whether the effects of exogenous cues in guiding attention differed across feature dimensions. Thus, we entered the cue effect data into a mixed ANOVA with cue type (6 exogenous cues) as a within-subject factor and group as a between-subject factor (Fig 4). The main effect of group was significant [$F(1, 38) = 5.65$, $p = .02$, $\eta_p^2 = .130$], with a larger mean cue effect in the TD group (443 ms) than in the ID group (294 ms). The main effect of cue type was also significant [$F(5, 190) = 2.99$, $p = .01$, $\eta_p^2 = .073$], but none of the pairwise comparisons survived Bonferroni-correction, indicating no significant difference between the six exogenous cue types ($t(39)$s $< 2.57$, $ps > .21$, $ds < .407$). The interaction between group and cue type was not significant ($F(5, 190) = 1.66$, $p = .15$, $\eta_p^2 = .039$), indicating that the pattern of the cue effect across exogenous cues was similar for both groups. These results reflect that the effect of

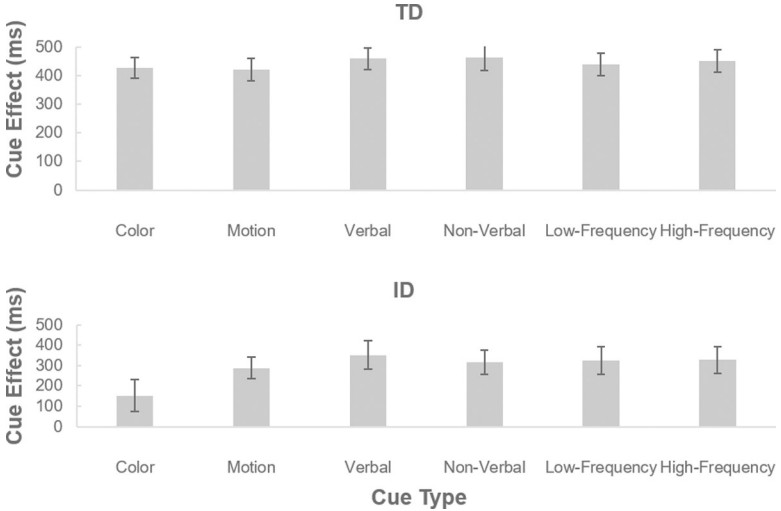

**Fig 4. Mean cue effect for each exogenous cue type, shown separately for the TD and ID groups.**

exogenous cues to guide attention was distributed evenly across feature dimensions for both people with ID and typically developing controls.

## The effect of repetition priming

The efficiency of history-driven attentional mechanism was examined by measuring the priming effect due to repetition of experimental conditions. We first divided trials into repeated and non-repeated subsets depending on whether the *cue type* on the current trial (n) is the same as that on the previous trial (n-1) or not, and observed whether there is improvement in search performance in repeated trials as compared to non-repeated trials. The purpose of this analysis was to examine whether repeated encounters of the same cue type has a positive effect on selecting and utilizing the cue information to deploy attention, regardless of the spatial location of the upcoming target. The IE data were entered into a mixed ANOVA with cue type repetition (repeated, non-repeated) as a within-subject factor and group (TD, ID) as a between-subject factor (Fig 5A). There was a significant interaction between cue type repetition and group [$F(1, 38) = 21.18$, $p < .001$, $\eta_p^2 = .358$], indicating that the pattern of cue type repetition priming was different between groups. Separate paired t-tests for the two groups revealed a significant repetition priming effect in the TD group, with better search performance in cue type-repeated trials (807 ms) than in non-repeated trials (867 ms) ($t(19) = 7.54$, $p < .001$, $d = 1.686$). On the contrary, the ID group showed a significant *decrease* in search efficiency in cue type-repeated trials (1652 ms) as compared to non-repeated trials (1551 ms) ($t(19) = 2.96$, $p = .008$, $d = .662$). Surprisingly, the repetition of the same cue type had an *adverse* effect on search efficiency in the ID group, which was opposite to the pattern observed in the TD group. These results suggest that the function of history-driven attentional mechanism that enhances the processing of repeated features of cue is not only deteriorated but even reversed in individuals with intellectual disability.

We also analyzed the effect of repetition priming of the *target location* (8 cube positions) and *target side* (left or right visual field), in the same manner as the cue type repetition. The repetition of target location had a significant main effect [$F(1, 38) = 5.54$, $p = .02$, $\eta_p^2 = .127$] with better search efficiency in target location-repeated trials (1171 ms) than in non-repeated trials (1217 ms), with no significant interaction between target location repetition and group [$F(1, 38)$

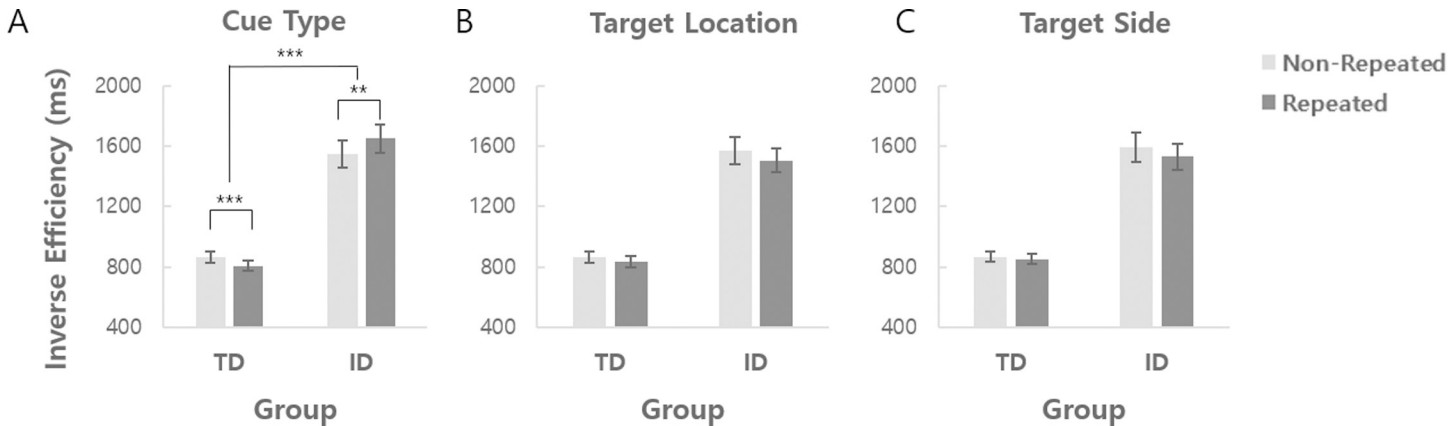

**Fig 5. The repetition priming effect for different conditions.** (A) Mean inverse efficiency for cue type non-repeated vs. repeated trials in the TD and ID groups. (B) Mean inverse efficiency for target location non-repeated vs. repeated trials in the TD and ID groups. (C) Mean inverse efficiency for target side non-repeated vs. repeated trials in the TD and ID groups. $^{**}p < .01$, $^{***}p < .001$.

= .90, $p$ = .350, $\eta_p^2$ = .023] (Fig 5B). This indicates that both typically developing controls and individuals with ID showed the target location repetition priming effect in a similar way. Consistently, search efficiency was marginally better in target side-repeated trials (1192 ms) than in non-repeated trials (1231 ms) [$F(1, 38)$ = 3.58, $p$ = .066, $\eta_p^2$ = .086], with no significant interaction between target side repetition and group [F(1, 38) = 1.46, $p$ = .234, $\eta_p^2$ = .037] (Fig 5C). These results suggest that the function of the history-driven attentional mechanism that guides selective attention to the previously attended *location in space* is relatively preserved in ID.

## The effect of target location and feedback

For additional exploratory analyses, we first examined how the location of the target influenced search performance by entering IE data into a mixed ANOVA with target location (8 cube positions) as a within subject factor and group as a between subject factor. There was a significant main effect of target location [$F_c(2.23, 84.86)$ = 6.07, $p$ = .002, $\eta_p^2$ = .138]. Post-hoc comparisons revealed that search was more efficient when the target appeared among the four central cubes (1146 ms) than among the four peripheral cubes (1296 ms) [$t(39)$ = 5.91, $p <$ .001, $d$ = .934], with no significant difference between cubes in the upper versus lower row [$t(39)$ = .94, $p$ = .71, $d$ = .148]. The interaction between target location and group was not significant [$F_c(2.23, 84.86)$ = 2.41, $p$ = .08, $\eta_p^2$ = .060]. These results indicate that both TD and ID groups processed targets presented in the central area more efficiently, with no particular attentional bias toward the upper or lower visual field. Similar analyses with target side (left, right visual field) and response hand (left, right) revealed no significant main effect or interaction (F(1, 38)s < 3.48, $ps$ >.07, $\eta_p^2$s < .084), confirming that both groups showed no particular attentional bias toward the left or right visual field, and no particular motor benefit for the left or right hand. In other words, their overt attention was mostly focused on the center of the screen with no particular shift in space as a baseline. These results provide evidence that presenting the target number on the center of the screen at the start of each trial was effective in guiding participants' attention to the center at first, and then the presentation of an informative spatial cue shifted participants' overt attention towards the corresponding side.

We also examined whether the presence and type of feedback (visual, auditory, tactile, and none) influenced search efficiency, by entering the IE data into a mixed ANOVA with

feedback block (visual, auditory, tactile, and none) as a within-subject factor and group as a between-subject factor. The results showed no significant main effect [$F(3, 114) = .64$, $p = .59$, $\eta_p^2 = .016$] or interaction [$F(3, 114) = .67$, $p = .57$, $\eta_p^2 = .017$] involving feedback block. This indicates that getting a different type of feedback (visual, auditory, tactile, and none) on accuracy of each response did not influence the overall search performance in our experimental task.

## Discussion

Successful achievement of most cognitive tasks in everyday life requires attention, the ability to enhance currently relevant information while inhibiting other sources of information. Therefore, elucidating the multifaceted attentional deficits in ID is critical in understanding, predicting, and improving their cognitive and behavioral performance. Theories of attention have demonstrated that there are top-down, bottom-up, and history-driven attentional mechanisms [6, 7, 24], controlled by partially-segregated networks of brain areas [9, 10, 25]. While the functional aspects of these attentional mechanisms have been extensively studied with typically-developing adults [8], few studies have examined how those attentional mechanisms are differentially affected by ID. Furthermore, the majority of previous studies focused on processing of visual stimuli in highly unnatural conditions, leaving unclear about the characteristics of multisensory processing in ID in a natural 3-D environment. Utilizing multisensory attentional cues in VR, the current study compared the efficiency of top-down, bottom-up, and history-driven attentional processing across different sensory modalities in ID.

The current study provided several important novel findings that elucidate the aspects of attentional deficits in ID. First, the overall attentional deficits were more pronounced in top-down rather than in bottom-up processing, but with different magnitudes of top-down deficits across sensory modalities. Participants with ID showed significantly smaller cue effects for endogenous cues than for exogenous cues, suggesting a diminished function of the top-down attentional system. This is consistent with the previous research that showed selective impairment in top-down attentional control in other neurodevelopmental conditions that belong to the broader category of developmental disabilities, such as ASD [17, 18] or ADHD [19], and confirms that deficits in top-down system is also observed in ID. This could indicate that developmental disabilities share at least partially overlapping mechanisms of attentional deficits. Although the comorbidity rates are very high among ID, ASD, and ADHD and many of their behavioral phenotypes are overlapping, it is important to note the etiological heterogeneity of the population and clarify the cognitive and behavioral profiles of each condition [49]. For clear identification of common and distinctive attentional characteristics in developmental disabilities, the behavioral performance of carefully controlled samples (e.g., mental age matched) from each condition should be directly compared within the same experimental paradigm in future studies. Interestingly, the relative impairment in top-down attentional control was more pronounced in auditory and tactile sensory modalities, rather than in the visual sensory modality. Considering that the experimental task in the current study required *visual* search of a specific target number, it is possible that the ability to utilize endogenous cues within the same sensory modality as the current task is relatively preserved, whereas *cross-modal* top-down attention is particularly impaired in ID. Alternatively, individuals with ID might rely more heavily on vision in general, regardless of the sensory modality required in the current task. To verify these hypotheses, it would be necessary to manipulate the sensory modality of attentional cues and that of the task fully crossed in future studies. There is also a possibility that the burden on working memory to maintain multiple associative rules for endogenous cues was too much for ID participants, causing less utilization of certain

endogenous cues. An experimental task with a fewer number of well-learned endogenous cues and a fixed search target to lower the burden on working memory would be able to clarify the reason behind the observed differences in cueing effects across sensory modalities.

Another potential hypothesis for relatively preserved endogenous cueing effects in visual modality in ID is that the central arrow worked as a hybrid cue, engaging both voluntary and reflexive attentional orienting. Previous studies showed that central symbolic cues that are highly overlearned or have social importance, such as arrows [50, 51], gaze direction [52], finger pointing [53], and words indicating a spatial direction [54] can orient attention in a reflexive manner even if they are not predictive of the target location. Ristic and Kingstone [55] also showed that predictive arrow cues engage *both* volitional and reflexive attention, with these effects combining in an interactive manner. Thus, the relatively preserved cueing effect for an arrow cue in ID might reflect the combined effect of top-down and bottom-up attentional mechanisms, rather than a pure top-down effect. In order to examine pure volitional attentional orienting, central cues should be truly symbolic, such as a color cue arbitrarily associated with different directions [56, 57]. In this study, however, we could not utilize such arbitrarily-learned symbols as endogenous cues, considering the impaired learning abilities in the ID group as compared to the TD group. Since the main purpose of our experiment was to measure the strength of attentional orienting, rather than the ability to learn and interpret the meaning of arbitrary symbols, we used a well-learned arrow cue that both ID and TD groups could intuitively utilize to orient attention. Auditory and tactile endogenous cues were also designed to make them conceptually similar to the arrow cue, utilizing the direction of the sound source movement or the vibration source movement. In future studies, arbitrary endogenous cues that are matched for intuitiveness across sensory modalities could be extensively trained on ID participants to examine the selective impairment in pure top-down attentional orienting across different sensory modalities. At the same time, it would be important to elucidate the specific cognitive process that caused impaired top-down orienting responses in ID: the overall diminished effect of endogenous cues could have been originated from impaired learning of associative rules, imprecise attentional template in working memory, reluctance on utilizing cue information for conservation of cognitive resources, or impaired function of deploying attentional resources. Verifying the extent of each cognitive process' involvement in producing impaired top-down orienting responses in ID would be an important topic for future studies.

Second, the function of the history-driven attentional mechanism was significantly altered in individuals with ID. Counterintuitively, the search performance of participants with ID was worse in trials where the same cue type was immediately repeated, showing a *reversed* repetition priming effect. This was opposite to the performance of the typically developing group, who showed significant benefit of repeated cue types. On the other hand, the repetition priming effects for the specific target location and target side were observed in both groups, suggesting that the function of the history-driven attentional mechanism that guides attention to the previously attended *location in space* was still preserved in ID. These seemingly contradicting results can be explained by considering that history-driven attentional guidance is closely intertwined with implicit learning, and implicit learning is not a unitary construct. Instead, distinct neural circuits are implicated in repetition priming, perceptual-motor procedural learning, and operant conditioning [58, 59]. For instance, Barnes et al. (2010) showed that ADHD children displayed atypical perceptual-motor sequence learning but intact contextual learning, and explained that ADHD is mediated by dysfunctional frontal-striatal-cerebellar circuits, which are involved in implicit learning of perceptual-motor sequences but not visual-spatial context. Thus, it is reasonable to assume that dysfunctional history-driven attentional guidance in ID is observed in some forms of implicit learning and not in others, based on the

etiology and the affected neural circuits. Consistent with our results that the repetition priming effect for locations in space was still preserved in ID, previous studies that involved implicit learning of *visual-spatial* information, such as contextual cueing [42] or repetition priming of visual perception [60], reported comparable facilitation effects in ID and TD individuals. On the other hand, when repetition priming for complex *verbal* material was investigated, a contradictory pattern of results emerged, with some reporting comparable priming effects for ID and TD groups [61], and others reporting reduced priming in ID individuals [62, 63]. It was also reported that ID children showed significantly impaired procedural learning than TD children [64]. Careful subtyping of the implicit learning process and concurrent utilization of neuroimaging techniques in future studies will shed light on the segmented function of the history-driven attentional mechanism in ID. It should also be noted that our results are not generalizable to all etiologies of ID, since it has been reported that ID individuals with different etiologies (e.g., Down syndrome, Williams syndrome) showed differential performance in an implicit learning task [65]. Finally, since the sample size of the current study was determined by a power analysis using parameters of the main analysis of interest, there is a possibility that the results from other analyses might be underpowered. Replications of the observed pattern of results with a larger sample size would fortify the argument for altered history-driven processing in ID.

The results of the current study not only contribute to understanding and predicting the pattern of attentional performance in ID, but also provide important insights in designing training programs for them. Recently, there have been increasing number of attempts in developing cognitive/behavioral training programs for people with intellectual and developmental disabilities, using multisensory stimuli in VR [36–38, 66, 67]. In order to make effective VR training programs, it would be crucial to understand the vulnerability in cognitive processes and sensitivity to different stimuli of the target population in an environment as consistent as the training context. The knowledge obtained from the current study can be utilized in designing VR training programs tailored to the attentional characteristics of ID. First, VR training programs for people with ID could be designed to contain exogenous, rather than endogenous, attentional cues that directly highlight the targeted location or object in space, in order to effectively guide the trainee's attention to the currently important area in the environment. Second, endogenous cues that are within the same sensory modality as the current task (e.g., arrow direction for visual search) would be more effective than cross-modal endogenous cues (e.g., sound direction for visual search) for people with ID. Third, rather than presenting the same cue type repeatedly, it would be better to diversify the type of attentional cues to effectively guide the trainee's attention. Additionally, the experimental paradigm we developed using VR HMDs has wide applicability to a broader population to investigate the individual or group profiles of cognitive processing of multisensory attentional cues.

Taken together, the current study provides a comprehensive picture of how top-down, bottom-up, and history-driven processing of multisensory attentional cues is affected by ID: The deficits in goal-driven attentional control are more pronounced than those in stimulus-driven attentional control, with different magnitudes of impairment across sensory modalities. Also, the effect of history-driven attentional guidance is diminished or even reversed for some type of repetitions. These results indicate that the impact of ID on attentional processing is not general, but specific to attentional mechanisms and sensory modalities.

## Supporting information

**S1 File. Supplementary analyses on the speed-accuracy tradeoff and log-transformed RTs.**
(DOCX)

## Acknowledgments

We thank the Daejeon vocational training center for persons with developmental disabilities for their cooperation in screening and recruiting participants with intellectual disability.

## Author Contributions

**Conceptualization:** Heesook Shin, Youn-Hee Gil, Jeongmi Lee.

**Data curation:** Jinwook Kim, Eugene Hwang, Jeongmi Lee.

**Formal analysis:** Eugene Hwang, Jeongmi Lee.

**Funding acquisition:** Heesook Shin, Youn-Hee Gil, Jeongmi Lee.

**Investigation:** Jinwook Kim, Eugene Hwang, Heesook Shin, Youn-Hee Gil, Jeongmi Lee.

**Methodology:** Jeongmi Lee.

**Project administration:** Heesook Shin, Youn-Hee Gil, Jeongmi Lee.

**Software:** Jinwook Kim.

**Supervision:** Jeongmi Lee.

**Validation:** Jeongmi Lee.

**Visualization:** Jinwook Kim, Eugene Hwang.

**Writing – original draft:** Jinwook Kim, Eugene Hwang.

**Writing – review & editing:** Jeongmi Lee.

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
