## [Decision Letter · Decision Letter 0]

14 Jul 2021

PONE-D-20-40451

Top-down, bottom-up, and history-driven processing of multisensory attentional cues in intellectual disability: An experimental study in virtual reality

PLOS ONE

Dear Dr. Lee,

Thank you for submitting your manuscript to PLOS ONE. Sorry for the relative delay in evaluating your manuscript. It was initially very hard to find reviewers for the paper as this required many iterations. Luckily, in the end, I was able to secure three expert reviewers, and I have read your paper myself. I would like to thank the reviewers for taking their time to provide a number of constructive feedback on the MS, and for taking this assignment even in this time in which most people are feeling overwhelmed with work and the global situation imposed by the pandemic.

That said, the reviewer's recommendations were very heterogeneous: from straight rejection, to major revision, to minor revision. My own reading aligns with several of the criticisms the reviewers raised, but I'd say that the paper has merits that could warrant its publication pending major revisions. Hence I am inviting the authors to submit a revision provided that they thoroughly address all the reviewer's comments, and some concerns I also have and that will be detailed below. Note that I will invite the reviewers again to assess if their concerns have been alleviated by the revisions made, so please make your best effort to address all of their comments.

I will not repeat all the points raised by the reviewers here. Instead, in addition to their points, I also had some comments I'd like to see addressed to make sure the paper is on its best shape possible for eventual publication.

First of all, I'd like to thank the authors for providing their data on the OSF. However, I should note that I could not find a data-book (which provides the meta-data information) associated with the data. Please add a file to the OSF with the meta-data: describe what is presented in each column of the data-file, the valid values within the column and their meaning (e.g., what the cue-code number means in terms of the exact type of cue presented?). The matlab script does not include this information as well. This is very important to ensure proper documentation and preservation of the data.

Please provide statistical justification for collapsing the data of the neutral cue conditions. Report an ANOVA and please provide the RT and accuracy means on a table for each cue type (this could also appear in a supplementary file).

I did not follow the description of the feedback manipulation. It is mentioned that there were trials with and without feedback, but then the authors state that there were 4 feedback blocks for a total of 200 trials. What about the no-feedback blocks? Are there no-feedback blocks? Are the data from these blocks being analyzed? In the end of the results section it is mentioned that there were some evaluation of the effect of feedback. It is not clear how this was done.

Overall, I’d caution the authors against manipulating so many variables in a single experiment in the future. The number of trials per design cell seems rather in the low end. I want the authors to report exactly how many trials per design cell they have. Overall, they should not report analyses for which they would have fewer than 20 responses per design cell, as this is likely to be unreliable. Besides that, the authors mention 10 types of cues, but I counted 12 types of cues. Are they already assuming that the neutral cues form a single condition in the break-out of the cues?

The use of inverse efficiency scores has been cautioned against in the literature (see for example Bruyer and Brysbaert, 2011) unless some conditions are met: relative high accuracy (which was the case here) but also absence of speed-accuracy tradeoffs. Can you please indicate whether the RTs and PE go in the same direction by presenting their correlation? Overall, it would be reassuring to see that analyses based only on RTs reproduce the same pattern of results. These supplementary analysis only on RT could be reported in supplementary file. In fact, given the much slower RTs of the ID group, it might be more appropriate to analyze logRTs. This would make sure that the interaction between group and cue is not due to the proportional slowing of the RTs in the ID group (similarly to the concerns raised in the aging literature). This is important, since this would make sure the interaction is not simply an artifact of the perceptual/motor-speed difference between groups, but indeed reflect an effect on attention guidance. If the authors present these analysis in the supplementary file, they should nevertheless indicate in the main text if the interaction was still significant in this analysis.

l.320-324: The separate ANOVAs do not show that the effect is smaller in one condition than the other. They show that there is a main effect of cue in both groups, which renders probable that the interaction is due to the smaller effect in the ID group.

l.345-346: The results of the separate ANOVAs are not being reported. Please report the actual statistics associated with each of them, instead of simply reporting it verbally. Besides that, it is not proper to only report the p-value for a comparison of cue types in the ID group. Report the full statistical analysis.

I take the point from Reviewer 1 that the arrow cues can be considered as an hybrid, hence the conclusion that endogenous cues in the visual modality is preserved should be made with caution and the relevant literature on the status of arrow cues should be reviewed.

Overall, please report the degrees of freedom for the all statistical tests, including the t-tests!

References  

Bruyer, R., & Brysbaert, M. (2011). Combining Speed and Accuracy in Cognitive Psychology: Is the Inverse Efficiency Score (IES) a Better Dependent Variable than the Mean Reaction Time (RT) and the Percentage Of Errors (PE)? Psychologica Belgica, 51(1), 5–13. https://doi.org/10.5334/pb-51-1-5

Thank you for submitting your paper to PLOS ONE.

We look forward to receiving your revised manuscript.

Kind regards,

Alessandra S. Souza, Ph.D.

Academic Editor

PLOS ONE

Journal Requirements:

Reviewers' comments:

Reviewer's Responses to Questions

**Comments to the Author**

1. Is the manuscript technically sound, and do the data support the conclusions?

Reviewer #1: No

Reviewer #2: Partly

Reviewer #3: Yes

2. Has the statistical analysis been performed appropriately and rigorously? 

Reviewer #1: No

Reviewer #2: Yes

Reviewer #3: Yes

3. Have the authors made all data underlying the findings in their manuscript fully available?

Reviewer #1: Yes

Reviewer #2: Yes

Reviewer #3: Yes

4. Is the manuscript presented in an intelligible fashion and written in standard English?

Reviewer #1: Yes

Reviewer #2: Yes

Reviewer #3: Yes

5. Review Comments to the Author

Reviewer #1: In several respects, this is an ambitious study. The authors implement a large number of different experiment conditions in an effort provide a comprehensive characterization of how attentional orienting is influenced by intellectual disability (ID). I appreciated the potential translational implications to which the authors draw reference. With that said, however, I think it is rather unclear what exactly we can conclude from this experiment. In fact, it would seem that the majority of the findings could be explained very simply by the ID participants having a more difficult time interpreting the meaning of artificial (endogenous cue) signals, which would be quite unsurprising in ID and is not helped by the other demands of the task (e.g., maintaining an unpredictable target digit in WM).

Specific Comments

1. It is unclear to what degree the less prominent orienting on the basis of endogenous cues in ID is due to a deficit in top-down attention or to an impoverished ability to apply task instruction more generally. As I understand it, the practice session ensures the ability to correctly report the target within the timeout limit, not to interpret the meaning of the cues. Particularly given the diversity of cue conditions, there is a lot of information for participants to track in this paradigm, and participants with ID may have simply been more likely to struggle understanding the rules concerning the meaning of the cues at all (hence the near-zero cuing effects in some cases) and/or lose track of these rules on a subset of trials. Were this the case, the data would seem hardly surprising and probably just a reflection of impaired cognition.

2. The endogenous visual cues are quite different from the other endogenous cues with respect to their intuitiveness. In fact, there is evidence that arrow cues can to some degree function as exogenous cues (and produce cuing effects even when non-predictive), likely due to their highly overlearned meaning (which contrasts with the other endogenous cues in this experiment). As such, the intact cuing effects for the "endogenous" visual cues in ID may simply reflect a substantially reduced role for task instruction producing those cuing effects (and thus be consistent with the possibility outlined in point 1). I would also strongly caution against drawing conclusions concerning a sparing of endogenous attention in the visual modality: it may well have nothing to do with the modality and everything to do with the specific choice of cues.

3. We do not know whether the "exogenous" cues were really effective as exogenous cues, since there are no invalid trials and the cues are 100% informative. These are really just more intuitive endogenous cues that may or may not have an exogenous component. As with the prior points, this really muddles the interpretation.

4. Why are the selection history effects with respect to the cue not broken down by cue type (modality and endogenous vs. exogenous)? Especially in the case of endogenous auditory and tactile cues, this really is not a fair comparison since those cues did not really show much of any cuing effect in ID (and more generally, if the ID participants struggled to use those cues, there is really not much in the way of selection that would repeat). And what about whether the direction of the cue repeated or changed (and not just the type of cue)? There is quite a bit going on here that could really change the interpretation. As things currently stand, I would strongly hesitate to conclude that selection history per se is fundamentally altered in ID without a more careful look at this data.

5. Given that this was a study on ID, I was a little surprised that the authors designed the task such that the target digit changed unpredictably trial-to-trial and required tracking. This demand may have made it more difficult for the ID participants to maintain the rules about how interpret the endogenous cues. More broadly, this manipulation could have been better justified.

6. For a patient study, the sample size seemed surprisingly small. There is no justification given for powering the study to detect a "medium" effect size, nor does the power analysis take into account the large number of conditions and the many analyses conducted. Also, for such a modest sample size, it is surprising that the samples were not more closely matched (esp. with respect to sex) -- given that the control sample is university students this should not be difficult to better match. With the low sample size and number of conditions, there really is not sufficient power to know whether an imbalance in participant characteristics is influencing the results at all, particularly as they might interact with ID.

7. Accuracy was very high in this task. It was therefore quite unexpected that the authors would focus their analyses on inverse efficiency (IE) rather than RT, since accuracy does not seem to be an informative measure in this situation and is likely adding noise to the data. This contrasts markedly from the kind of situations for which IE was developed, where both accuracy and RT are clearly influenced.

8. Why were eye and/or head movements not analyzed and reported? As a study on attentional cuing, these measures could be more informative than RT/IE.

9. The somewhat dramatic difference in mean RT between groups could be inducing some sort of ceiling effect in the ID group. More should be done to try to rule this out with the data.

Minor:

1. Although I am skeptical the IE is really the most appropriate dependent measure for this study (see above), I would note that IE should not be expressed in ms. It is really an arbitrary unit that is no more linked to ms than it is to proportion correct.

2. Some of the ANOVAs have overlap in the effects reported, with some identified effects to some degree restating each other (e.g., multiple main effects of group). I am not sure that this is inherently a problem, although the authors might be a little more careful in how they emphasize/contextualize different effects when this happens (make sure the reader understands the redundancy and that this does not constitute converging evidence).

3. "highly unnatural conditions" (line 462). It is not clear to me to what degree the present study really marks an improvement in this respect. Yes, it is in a 3-D rendered environment, but the task is still quite a bit artificial (in a variety of ways).

Reviewer #2: This paper examined the attentional mechanisms in adults with intellectual disabilities vs. typically developing controls (N = 40). The work reported is important and well grounded (and authors made that clear). Writing is very good in terms of coherence and clarity (some proofreading may be needed just to grasp some minor issues, sometimes the use of articles seemed to be missing). My only concern, easily solved through re-writing, is the lack of clear research questions and hypotheses, explicitly matched with the data analytic plan (which could be presented in a more systematic way, rather than “as we read”). This is further detailed below, along with some minor suggestions for authors to consider, which may help them to further improve their work.

1. In the introduction I felt the need of some more (not too much) information about the intellectual disability (ID) as a neurodevelopmental disorder.

2. On p. 4, starting on line 85, authors are introducing a third mechanism, but this only becomes clear in the middle of the paragraph. In the beginning, readers are kind of lost trying to get why that info is relevant. Making it clear up front may help understanding.

3. On p. 5 the starting of the paragraph with “another issue” seems also a bit disconnected (something like “by the way”). It’s a minor aspect but I’d like to challenge authors to try a better transition from the presentation of the mechanisms to the modality/artificial conditions issue.

4. In the last paragraph of p. 6, when the study is presented, authors refer to “issues”. I believe it would be better to present research questions and hypotheses (though a general hypothesis is presented for issue #1, it was not for issue #2).

5. On top of p. 7, authors say “with two different feature dimensions in each modality (e.g., color and motion features for visual modality)”. The purposes of these two dimensions (for visual and the other modalities) is not clear.

6. Next authors refer to “compare behavioral performance across conditions”. It would be helpful to clarify exactly which kind of behavioral performance. This may also help to understand what is meant by “efficiency”.

7. I’d like to praise authors for including power analyses. However, as several analyses were done, I was not sure about what was used (mainly in terms of the within subject factor). Also, authors refer to the interactions, but what about the main effects? Authors could additionally include a citation to support the decision of the effect size used as reference.

8. Authors used inferential statistics to compare age between groups. Any reason for not comparing at least gender and handedness? Looking at the frequencies, they seem similar, but a statistical test would give more confidence on the groups equivalence.

9. On p. 9, line 175 it is said that participants with clinical history of other neurodevelopmental problems (e.g., Down syndrome) were excluded. But looking at the table, we can see those with ASD were kept. Any reason for this double criterium?

10. Concerning the other reason for excluding “any other behavioral problems”, not clear what this means exactly (maybe adding an example) and how was done. An indication of how many participants were excluded would be helpful as well as making it clear that the 40 were after exclusions.

11. As far as I understood, in the experimental task, authors manipulated cue and feedback, but also cue feature dimensions. Maybe that could be said up front before presenting the different manipulations. In particular, the feedback part, I just got the whole point of it after reading the whole paragraph.

12. On a related vein, though it may be there implicitly, I think authors should establish a clear link between the theory/hypotheses and the conditions. For example, the endogenous/exogenous and modality parts are quite obvious, but the role of the feedback and the dimensions (color/motion, etc.) was not.

13. The description of the experimental task is clear. I did however miss an explicit indication about the measures extracted from the task (I believe this point relates with the comments above on behavioral performance and efficiency). This may also help to understand the exact meaning of “inversed efficiency”. The formula is clear but the meaning not so.

14. Still about the analysis, authors mentioned a mixed ANOVA but just present the between subject factor. A clear presentation of the factors seems needed.

15. Perhaps is a question of writing, but I didn’t get why main effects and interactions were followed with separate analyses (which may increase Type error I rate) instead of pairwise comparisons or post-hocs for main effects and simple effects analyses for interactions, within the ANOVA (and not with additional tests). The result might be the same, but the approach is much more parsimonious and reduces Type error I. An additional advantage is that, in the interactions, authors will be able to compare not only condition differences separately by group but also group differences separately by conditions (this latter part was not examined, and may help to understand findings).

16. Despite that section explaining the analyses before the results, when we go through that section, we realize that other analyses were conducted, with varying within subject factors, and sometimes with unclear purposes. I understand this might be the “response” for the comment #14 above, but it is a bit confusing and hard to follow. The analyses should be all explained before the results section.

17. If authors used the inversed efficiency score, which combines accuracy and RT, why it is said on p. 15 lines 301-302 that only RT for correct trials were analyzed? This would mean that for compute the inversed efficiency score, the accuracy one would be a constant.

18. Through the results section I believe P should be changed to p (lowercase).

19. Overall, the results section is not aligned with the present study description and is a bit hard to follow, mainly because readers don’t have a clear idea about what research question is each analysis responding to, and also because authors mixed explanation for the analyses with results and discussion statements.

20. On p. 18 lines 374 authors mention “six feature dimensions, namely color, motion, verbal, nonverbal, high-frequency, and low-frequency”. This is mentioned briefly in the method but the purpose of comparing this was not clear. This happens with several other analyses (e.g., target side/location, feedback, etc.)

21. My main concern with this ms. is that there is not clear match between the research questions, how each one was methodologically addressed, what are the hypotheses for each one, and which analyses were used to test each and every one of them. Only in the results section, do we realize there are some “new” specific research goals, not fully introduced in the lit review.

22. This match between RQ/hypotheses/analyses will also help to organize the discussion. As it is (mainly the first/second paragraphs) is clear and insightful, but I believe it could be expanded. This could be done by addressing the results of the analyses in full (now, some of them were disregarded, e.g., on feedback etc.) as well as by going beyond a simple summary of the findings (e.g., p. 24, lines 484-491) and interpreting them.

23. I’d also like to read something more about the implications of the findings for theory. The introduction is strongly framed to claim on the novelty of the study and on the overcoming gaps in the literature. But then the discussion is more “modest” and does not establish that link back to theory.

24. Some indications for future research would also be useful.

Reviewer #3: The current study examines attentional abilities in adults with and without Intellectual Disability and demonstrates consistent impairments in multiple types of attention but with the greatest impairment in top-down, cross-modal attention guidance. Overall, it is a well-written article that uses a carefully thought out attentional paradigm in a novel virtual reality setting to better understand attentional functioning in this population. Assessing attentional functioning in this population addresses a gap in the literature, which has often failed to consider the possibility of preserved functioning for some types of attention. The sample is small, and despite the power analysis, requires some additional discussion of generalizability given the heterogenous nature of the population of interest. Additional details about this sample and additional explanation of the potential theoretical basis for some findings will strengthen the manuscript. Specific comments include:

1) I appreciate that the authors provide information about the severity of intellectual disability in their sample overall, however given substantial heterogeneity in this population additional descriptive information would be helpful in integrating their work with the field. Are IQ scores available? Also scores on any standardized measures of adaptive functioning? Is there any additional standardization of diagnosis or any specific medical or other comorbidity rule-outs for participants in this sample?

2) In several instances authors seem to lump ID with ADHD and ASD (e.g., in justifying why top-down attentional problems may be present). While these are all developmental disabilities and ID may occur in higher rates in ADHD and ASD, they are also fundamentally different populations. Indeed many individuals with ASD and ADHD have normal or above average IQ and adaptive skills. This should be clarified throughout the manuscript. Additional discussion related to whether we would expect the same mechanisms of attentional problems in these populations or might expect different attentional patterns (e.g., more general deficits in one group versus more specific in another diagnositc group) will add important nuance to the current discussion.

3) The authors find an interesting pattern in which the history-driven attentional guidance may actually operate in a counter-productive (and counterintuitive) manner in those with ID. Is there any prior research or theory that might help explain this finding? Some additional discussion of how the patterns map onto prior work in ID, known differentiation of networks supporting history-driven attentional guidance, or other possible theoretical explanations for this finding will help readers understand whether this pattern makes sense or is more likely to be a fluke of current data.

4) Additional discussion of generalizability of findings should be included.

6. PLOS authors have the option to publish the peer review history of their article (what does this mean?). If published, this will include your full peer review and any attached files.

Reviewer #1: No

Reviewer #2: No

Reviewer #3: No

---

## [Author Response · Author response to Decision Letter 0]

3 Sep 2021

Point-by-point responses to the academic editor and reviewers' comments are in the "Response to reviewers" file.

---

## [Decision Letter · Decision Letter 1]

19 Oct 2021

PONE-D-20-40451R1Top-down, bottom-up, and history-driven processing of multisensory attentional cues in intellectual disability: An experimental study in virtual realityPLOS ONE

Dear Dr. Lee,

Thank you for submitting your revised manuscript to PLOS ONE. I have sent your manuscript to the reviewers of the original submission. They were very positive towards the changes implemented. Nevertheless, Reviewer 1 still makes the argument that your presentation of the results, and in some case of design choices, could be more nuanced and with a stronger discussion of caveats. Reviewer 2 has just some minor final suggestions for improvements. Hence I am inviting you to submit a final revision of your paper in which you address these last points. Please do your best job to clarify them. I hope to make a final decision on your manuscript on this next round. Indicate clearly how you changed the MS in response to each of the comments thereby improving its clarity. In addition, submit a response letter indicating your rationale for addressing each comment in the manner you did. Thank you for considering PLOS ONE as an outlet for your research.

We look forward to receiving your revised manuscript.

Kind regards,

Alessandra S. Souza, Ph.D.

Academic Editor

PLOS ONE

Journal Requirements:

Reviewers' comments:

Reviewer's Responses to Questions

**Comments to the Author**

1. If the authors have adequately addressed your comments raised in a previous round of review and you feel that this manuscript is now acceptable for publication, you may indicate that here to bypass the “Comments to the Author” section, enter your conflict of interest statement in the “Confidential to Editor” section, and submit your "Accept" recommendation.

Reviewer #1: (No Response)

Reviewer #2: (No Response)

2. Is the manuscript technically sound, and do the data support the conclusions?

Reviewer #1: Partly

Reviewer #2: Yes

3. Has the statistical analysis been performed appropriately and rigorously? 

Reviewer #1: Yes

Reviewer #2: Yes

4. Have the authors made all data underlying the findings in their manuscript fully available?

Reviewer #1: Yes

Reviewer #2: Yes

5. Is the manuscript presented in an intelligible fashion and written in standard English?

Reviewer #1: Yes

Reviewer #2: Yes

6. Review Comments to the Author

Reviewer #1: The authors have made changes that have resulted in some improvement to the manuscript, but the authors have responded to several of my prior points in the response letter in a manner that I did not find convincing, in some cases in whole and in some cases in part. Below I refer to my prior points and offer some follow-up:

Previous point 1: If they showed such prominent orienting during practice, it is puzzling that they showed no cuing effect for those non-visual endogenous cues in the main experiment. It is also unclear how long these rules were retained. Given the obvious difference in the need to retain an active set of rules in WM for the endogenous (non-visual) cues and the difficulty that ID participants would be expected to have with this by definition, I found this initial practice session an underwhelming basis upon which to completely rule out the idea that the results could be at least partly explained by an impaired ability to closely follow task rules. I think a sterner caveat would be appropriate here.

Previous point 2: I appreciate those additions to the text, but in the abstract and elsewhere (e.g., Results), the authors still make the claim that how ID affects attention is specific to certain sensory modalities which may not be the case. I would recommend more carefully tempering those claims (especially in the abstract, which lacks this important context).

Previous point 3: I still do not understand how a 100% valid cue can be called an exogenous cue (regardless of whether it is compared to a neutral cue). The cue is highly informative. I think the authors need to be more cautious with the terminology they use here: many readers will see "exogenous" and assume reflexive orienting to a non-predictive cue, and without the cue being non-predictive, there is no way to know just how exogenous the associated orienting response is. That the authors call these exogenous cues is really just an assumption, and although I agree that these cues likely have some exogenous component, for all we know the cueing effects associated with them could be largely dominated by endogenous processes (calling into question that more-or-less categorical distinction forwarded in the paper and reified in the labeling of the task conditions). Indeed, they are just as informative as the endogenous cues (100%). I realize it is convenient to have an endogenous/exogenous factor and label it as such, but I do not think that is appropriate here.

Previous point 4: I appreciate that there are not many trials per cell, but I am not sure what exactly this analysis gets at without going into greater depth. What exactly does a mere repetition of cue type (collapsed across all other factors) tell you about selection history? And I understand that the "examination of the repetition priming effect was independent from the cueing effect analyses," but I still do not see how the lack of cueing effect does not call the meaningfulness of this analysis into question to some degree. If there is no evidence that ID participants "selected" the cue in some cases, then how can this analysis speak to "selection history" in those cases? What is the attentional process that the authors hypothesize is repeating in this case that could be blunted or accentuated by ID?

Previous point 5: I understand that the WM burden of keeping the digit in active memory is in and of itself low, but this is coupled with the need to keep the rules for interpreting the different cues in active memory as well (which I assume was not the case in those Woodman et al. and like studies), and this just adds one more thing to remember that may be more likely to push ID participants over the edge of what they can keep in active memory. The concern is not that ID participants forget the target digit, which I agree would be inconsistent with measured accuracy, but rather that the combined demand caused some ID participants to forget the rules for interpreting the cue (much more so than control participants), leading to the lack of cueing effects in some cases. I still think this is a weakness of the paradigm that requires more careful consideration.

Previous point 6: With such a small sample size, it is not much to show that the samples did not significantly differ on characteristics like gender. Only a very large imbalance will show up as significant. There are literally almost twice as many females in the TD group (!!), and this could be at least to some degree influencing the results. The TD sample is just university students, so I still do not understand why the samples were not more closely matched, especially with respect to gender. This at least deserves a sterner caveat.

Reviewer #2: Authors did a very good job in reviewing the manuscript. At this point, I just have three additional suggestions to improve the ms.:

1. The exploratory research questions should be stated (and justified) in the last paragraph of the introduction;

2. Effect sizes (including for non-significant results) should be presented for all analyses, including t tests;

3. If sample size was based on the main RQ, it looks that some exploratory RQ are underpowered. Either being or not the case, this issue should be discussed in the manuscript.

As a side note, the submission guidelines present lowercase p-values: “P-values. Report exact p-values for all values greater than or equal to 0.001. P-values less than 0.001 may be expressed as p < 0.001, or as exponentials in studies of genetic associations.” (https://journals.plos.org/plosone/s/submission-guidelines.#loc-statistical-reporting).

7. PLOS authors have the option to publish the peer review history of their article (what does this mean?). If published, this will include your full peer review and any attached files.

Reviewer #1: No

Reviewer #2: No

---

## [Author Response · Author response to Decision Letter 1]

19 Nov 2021

Please refer to the "Response to reviewers" file.

---

## [Editor Report · Decision Letter 2]

1 Dec 2021

Top-down, bottom-up, and history-driven processing of multisensory attentional cues in intellectual disability: An experimental study in virtual reality

PONE-D-20-40451R2

Dear Dr. Lee,

We’re pleased to inform you that your manuscript has been judged scientifically suitable for publication and will be formally accepted for publication once it meets all outstanding technical requirements.

Kind regards,

Alessandra S. Souza, Ph.D.

Academic Editor

PLOS ONE
---

## [Editor Report · Acceptance letter]

7 Dec 2021

PONE-D-20-40451R2 

Top-down, bottom-up, and history-driven processing of multisensory attentional cues in intellectual disability: An experimental study in virtual reality 

Dear Dr. Lee:

I'm pleased to inform you that your manuscript has been deemed suitable for publication in PLOS ONE. Congratulations! Your manuscript is now with our production department. 

Kind regards, 

on behalf of

Dr. Alessandra S. Souza 

Academic Editor

PLOS ONE